# Extranodal NK/T-Cell Lymphoma, Nasal Type: Genetic, Biologic, and Clinical Aspects with a Central Focus on Epstein–Barr Virus Relation

**DOI:** 10.3390/microorganisms9071381

**Published:** 2021-06-25

**Authors:** Miki Takahara, Takumi Kumai, Kan Kishibe, Toshihiro Nagato, Yasuaki Harabuchi

**Affiliations:** 1Department of Otolaryngology-Head and Neck Surgery, Asahikawa Medical University, Asahikawa 078-8510, Japan; t-kumai@asahikawa-med.ac.jp (T.K.); kkisibe@asahikawa-med.ac.jp (K.K.); hyasu@asahikawa-med.ac.jp (Y.H.); 2Department of Innovative Head & Neck Cancer Research and Treatment (IHNCRT), Asahikawa Medical University, Asahikawa 078-8510, Japan; 3Department of Pathology, Asahikawa Medical University, Asahikawa 078-8510, Japan; rijun@asahikawa-med.ac.jp

**Keywords:** extranodal NK/T-cell lymphoma, nasal type, Epstein–Barr virus (EBV), EBV-encoded small nuclear early region (EBER)-1, latent membrane protein (LMP) 1, EBV DNA, MPVIC-P

## Abstract

Extranodal NK/T-Cell Lymphoma, nasal type (ENKTL-NT) has some salient aspects. The lymphoma is commonly seen in Eastern Asia, has progressive necrotic lesions in the nasal cavity, makes midfacial destructive lesions, and shows poor prognosis. The lymphoma cell is originated from either NK- or γδ T-cells, which express CD56. Since the authors first demonstrated the existence of Epstein–Barr virus (EBV) DNA and EBV oncogenic proteins in lymphoma cells, ENKTL-NT has been recognized as an EBV-associated malignancy. Because the angiocentric and polymorphous lymphoma cells are mixed with inflammatory cells on a necrotic background, the diagnosis of ENKTL-NT requires CD56 immunostaining and EBER in situ hybridization. In addition, serum the EBV DNA level is useful for the diagnosis and monitoring of ENKTL-NT. Although ENKTL-NT is refractory lymphoma, the prognosis is improved by the development of therapies such as concomitant chemoradiotherapy. The basic research reveals that a wide variety of intracellular/cell surface molecules, cytokines, chemokines, and micro RNAs are involved in lymphomagenesis, and some of them are related to EBV. Understanding lymphoma behavior introduces new therapeutic strategies, such as the usage of immune checkpoint inhibitors, peptide vaccines, and molecular targeting therapy. This review addresses recent advances in basic and clinical aspects of ENKTL-NT, especially its relation to EBV features.

## 1. Introduction

Extranodal NK/T-cell lymphoma, nasal type (ENKTL-NT) has some salient aspects. Patients with ENKTL-NT are commonly seen in Eastern Asia [1,2,3,4] and Latin America [4,5] but less frequently in the United States and Europe [6,7,8]. ENKTL-NT usually develops progressive necrotic granulation in the nasal cavity and shows a poor prognosis [2,9]. Histologically, ENKTL-NT is composed of angiocentric and polymorphous lymphoreticular infiltrate, previously called “polymorphic reticulosis” [10,11]. Original cells of ENKTL-NT are either NK- or γδ T-cells, both of which express CD56 [2,8,12,13,14,15,16]. In 1990, the authors first discovered the presence of Epstein–Barr virus (EBV) DNA and protein in the cells [1], and then found the EBV latency pattern and the clonality of the EBV genome [2,16,17]. In the present day, ENKTL-NT is recognized as EBV-related lymphoma [16]. This characteristic is used for diagnosis and monitoring after therapy for ENKTL-NT. For instance, the in situ hybridization of the EBV-encoded small nuclear early region (EBER)-1 is necessary for the pathological diagnosis of ENKTL-NT [18], and circulating EBV DNA levels are of great utility as highly sensitive tumor markers [19]. Although the prognosis of ENKTL-NT is poor, recent advancements in therapies, including concurrent chemoradiotherapy [20] and intra-arterial infusion chemoradiotherapy [21], improved the outcome of ENKTL-NT, especially in the early stages. Since two cell lines were established from ENKTL-NT tissues [13], basic research of ENKTL-NT has developed rapidly, and now we know that various intracellular/cell surface molecule, cytokines, chemokines, and micro RNAs are involved in the development of ENKTL-NT. More importantly, some growth-related factors could be a target of further therapies. Because EBV affects the development of ENKTL-NT via these factors, EBV proteins, such as latent membrane protein (LMP) 1, will be the target also. In this review article, the authors introduce the current understandings of ENKTL-NT with a particular focus on its relation to EBV.

## 2. Historical Context

In 1897, McBride et al. [22] first mentioned a disease condition of progressing necrotic granuloma in the nasal cavity resulting in a rapid invasion of the nose and face (midline). The patients followed an aggressive and lethal clinical course, and the disease was initially named “lethal midline granuloma” from the clinical characteristics [23,24]. Conversely, based on the pathological characteristics, the disease was called “polymorphic reticulosis” [11] or “angiocentric lymphoma” [10], which was characterized by diffuse infiltrates of pleomorphic and atypical large and small lymphoid cells with frequent mitosis admixing with a large number of inflammatory cells, such as granulocytes, macrophages, and plasma cells, with ischemic necrosis and angiocentric and/or angioinvasive infiltrates in the lesions. In the late 20th century, phenotypic studies [9,25,26] revealed that the tumor cells had NK-cell (CD56) and T-cell (CD3) markers. Accordingly, this lymphoma was identified as ENKTL-NT [25].

Etiologically, we [1] first reported the presence of EBV DNA and EBV-determined nuclear antigen 1 (EBNA1) in the lymphoma cells from 5 Japanese patients. EBV involvement was subsequently reported in China [27], the United States [28], France [8], and other western countries [29]. Thus, ENKTL-NT is now recognized as an EBV-associated malignancy [16].

## 3. Epidemiology

There is a clear regional difference in ENKTL-NT prevalence. In Asia and South America, ENKTL-NT makes up 3–10% of non-Hodgkin lymphomas, whereas it makes up less than 1% in Western countries [30,31,32,33]. In addition, in the United States, it is estimated that ENKTL-NT represents approximately 1–2% of all T/NK-cell lymphomas and approximately 0.2% of all non-Hodgkin lymphomas [34]. The reason for the difference may be explained by a difference in race. The mongoloid race, which is popular in Asia and South America, may have the susceptibility gene of ENKTL-NT. Although the susceptible gene has not been discovered yet, relations to HLA loci have been reported in other EBV-associated malignancies, of which prevalence varies by region. For instance, HLA-A0207, which is common among Chinese people, is consistently associated with nasopharyngeal carcinoma (NPC) in Taiwan [35]. Moreover, meta-analyses in 13 published studies showed positive associations between NPC and the HLA-A2, B14, and B46 [36]. In EBV-positive Hodgkin lymphoma, HLA-A1 is reported to increase, and HLA-A2 decreases the risk of development [37].

Another explanation of the difference could be a relation to EBV. The existence of specific EBV strains or variants that electively infect NK/T-cells and stay in the cells by evading immunological surveillance may cause geographical distributions. However, there are few reports showing relationship-specific strain variants or mutants along with ENKTL-NT [18]. Unlike type 1 EBV, type 2 EBV can infect T-cells more effectively [38]; however, type 2 is not popular in East Asia, where T-/NK-cell malignancies are commonly seen [39]. In a previous study, we [40,41] showed that the EBV gene extracted from ENKTL-NT tissues had several missense mutations of amino acids recognized by CD8^+^ T-cells. Therefore, lymphoma cells infected with the mutated EBV may escape from cytolysis by immune cells.

Environmental factors are also possibly involved in the difference. Our group [42] has shown that exposure to pesticides and chemical solvents increased incidences of ENKTL-NT. In addition, Kojya et al. [43] reported familial ENKTL-NT with exposure to pesticides.

## 4. Clinical Features

The literature-based clinicopathological features are summarized in Table 1 [3,18,44,45,46,47]. The peak of the incidences of lymphoma occurs at the middle age of 40–50 years old, and the male-to-female ratio is approximately 2:1. More than 70% of the patients are diagnosed at an early stage, based on the Ann Arbor classification. The lesion is initially found as a necrotic granuloma and ulceration in the nasal cavity, which is an extranodal site [1,2] (Figure 1a,b). The tumor easily invades surrounding tissues, including the palate (Figure 1c), the nasopharynx (Figure 1d), facial skin (Figure 1e,f), the paranasal sinus, and the orbits, resulting in the destruction of the midline facial structure [1,22,24].

The most frequent symptoms are nasal obstruction and bloody nasal discharge [2,18,45]. In addition, prolonged fevers, as systemic symptoms, are also commonly seen [2,45,48]. In our study with 62 patients with ENKTL-NT [18], nasal obstruction, bloody nasal discharge, and prolonged fevers were seen in 49 (70%), 29 (47%), and 32 (52%) patients, respectively (Table 1).

## 5. Pathology

Pathologic findings of ENKTL-NT were known as diffuse infiltration of pleomorphic large and small lymphoma cells (Figure 2a), mixed with various inflammatory cells on the necrotic background [2,10,11]. The lymphoma cells express T-cell markers, such as cytoplasmic CD3 (CD3ε), as well as NK-cell marker CD56 (Figure 2b) [2,8,9,12,13,14,15,16]. Perforin, Fas ligand, and intercellular adhesion molecule-1 (ICAM-1) are also expressed in the ENKTL-NT cells [49].

In regard to the original cells of ENKTL-NT, there are two lineages: NK and T-cell [14,15]. We [2] and other groups [50,51] showed that T-cell receptor (TCR) gene rearrangement was proved in ENKTL-NT tissues in some patients. According to our investigation, ENKTL-NT tissue from 12 (35%) out of 34 patients had TCR gene rearrangement [18]. In fact, Nagata et al. [13] reported that two ENKTL-NT cell lines established from the tissues have different lineages: NK and T-cell.

## 6. EBV Status

In 1990, we first showed the presence of EBV DNA and EBV nuclear antigen (EBNA1) in the lymphoma cells from 5 patients with ENKTL-NT [1] (Figure 2c,d). According to the knowledge of that time, EBV-related lymphomas were of B-cell lineage, such as Burkitt lymphoma. Therefore, the surprising result that EBV could infect NK or T-cells propounded a new concept of EBV-related malignancies. In the tissue section, the cells stained by EBNA1 were also stained with CD2, suggesting that EBV-infected cells were not surrounding B-cells (Figure 2d). After this report, the same findings were reported from Chinese and Western countries [8,27,28,29]. Our investigation, using in situ hybridization, revealed that EBER-1, which was generally expressed in cells infected with EBV, was detected in the section in 59 (95%) out of 62 patients with ENKTL-NT [18] (Figure 2f, Table 1). These results suggest that ENKTL-NT is an EBV-related lymphoma.

Proof of the clonotypic EBV genome [2] is important for the verification that EBV is infected with the original cells before tumorigenic transformation. The lymphoma cells express EBNA 1 (Figure 2d) and LMP1(Figure 2e); this indicates that ENKTL-NT is categorized as a type II latency infection of EBV [2,52]. In regard to the expression of LMP1, we found LMP 1 mRNA in the tissues of all examined patients with ENKTL-NT, but the LMP1 protein in only half of the patients. This discrepancy may be explained by the methylation of LMP coding sequences [2,17,48]. Additionally, ENKTL-NT cells expressed LMP1 protein by immunohistochemical staining in 25 (47%) out of 53 patients with ENKTL-NT (Figure 2e, Table 1) [18].

In regard to the characteristics of EBV infecting the ENKTL-NT cells, the 30-bp deletion in the codon 343–352 of LMP1, which is famous for the B95-8 strain, was detected in the tissues of the vast majority of the patients [41,53]. Moreover, we detected several missense mutations in the epitope of the LMPs recognized by HLA-A2-restricted CTL [40,41]. Therefore, the infected EBV may have an ability to escape immune surveillance by CTLs, resulting in playing a role in lymphomagenesis.

## 7. EBV Infection of T or NK Cells

In primary infection, EBV in saliva infects B-cells by binding gp350/220 to CD21 and gH/gL/gp42 to HLA class II molecules [54]. Because both CD21 and HLA class II are expressed on the surface of B cells, B cells are the main hosts of EBV. Moreover, EBV could also infect the mucosal epithelium of the oropharynx [55]. This infection requires attachments between BMRF2 or the gH/gL of EBV and integrins on the epithelial cells [56].

In regard to NK/T-cells, there are some reports showing that EBV infects them. EBER-positive NK/T-cells were found in tonsils with acute infectious mononucleosis [57,58]. However, the mechanism with which EBV infects NK/T-cells remains unclear. Basically, neither CD21 nor HLA class II express in NK/T-cells. On the other hand, NK/T-cells express some integrins, which may act as the EBV-receptors in NK/T-cells as well as epithelial cells [39]. Because CD21 is expressed in premature T-cells and common lymphoid progenitors [59], EBV can infect premature T-cells during intra-thymic maturation and human immature T-cell lines in vitro [60,61]. Interestingly, the fact that the infection of both CD4^+^ and CD8^+^ T-cells [62], as well as both T-cells and NK-cells [63], were seen in patients with chronic active EBV infection (CAEBV) supports the hypothesis that EBV may infect premature T-cells. More recently, Smith et al. [64] reported that CD21 was detected in mature peripheral CD3^+^ T-cells by the anti-CD21 monoclonal antibody clone HB5 instead of the popular antibody clone Bly4. Moreover, a second minor strain of EBV, EBV type 2, could infect mature peripheral T-cells through HB5 antibody-detected CD21 [64].

Another possible mechanism is that B cells or epithelial cells infected with EBV may contribute to the infection of NK/T-cells in a cell-to-cell contact manner. NK cells attacked autologous EBV-infected B cells, activated, and acquired the B-cell membrane, including CD21 molecules, by synaptic transfer [65]. Interestingly, the transferred CD21 has the capacity to catch a viral particle of EBV on the NK cell. However, Lee et al. [66] reported that EBV failed to infect an EBV-negative NK cell line through synaptic transferred CD21. They also showed that EBV genes but not RNA were detected in the NK cells by only treatment of the EB-viral supernatant. This condition is known as latency stage 0, in which viral gene expression is mostly suppressed [67]. Along with B-cells primarily infected with EBV, the condition may be necessary for the avoidance of immune surveillance and the persistence of EBV in NK cells.

## 8. Gene Mutations

Genetic alternations, which also have pathogenic importance, have been reported in ENKTL-NT. For instance, deletion of the chromosome 6q21–25 was frequently seen in lymphoma tissues [68,69,70,71]. Gene mutations of apoptosis-related cell surface receptor Fas (Apo-1/CD95) were detected in the tissues of more than half of the patients [72,73]. Our group and others examined the mutations of major tumor-related genes, such as p53, K-ras, and c-kit [48,74,75,76,77]. According to the results, the p53 mutations were detected in 20–50% of patients; however, the K-ras, c-kit, and β-catenin mutations were rare. In addition, we [48] found that the p53 missense mutation was a factor that could predict poor survival.

## 9. Diagnosis by Using EBV Infection

It is difficult to diagnose ENKTL-NT by pathological examination with standard staining, such as hematoxylin-eosin, because of numerous necrotic backgrounds and mixtures of inflammatory cells [2]. Therefore, additional immunostaining, such as CD2, cytoplasmic CD3ε, CD56, cytotoxic molecules (perforin, granzyme B, and T-cell intracellular antigen 1) is needed for the diagnosis [2,18,78]. Of these molecular markers, CD56 is the most trustable marker for diagnosis because of a high positivity rate in the patients with ENKTL-NT (Table 1) [3,18,44,45,46,47].

Another promising method for diagnosis of the tissue sections is EBER in-situ hybridization. In the pathology laboratory of a general hospital, this procedure must be done by using commercial detection kits, which can be used for clinical diagnosis. Moreover, clear staining is usually obtained for the sections compared to immunohistochemical staining. The absence of EBV in the lymphoma cells excludes the diagnosis of ENKTL-NT according to a high positivity rate in the patients with ENKTL-NT (Table 1) [3,18,44,45,46,47]. Therefore, at least CD56 immunostaining and EBER in situ hybridization are required for diagnosis.

The differential diagnosis of ENKTL-NT can be judged by the common location of extranodal sites, especially the nasal cavity, the presence of EBER and CD56, and an elevated EBV DNA level, which will be mentioned later. It should be distinguished from other NK/T-cell lymphoproliferative diseases, such as blastoid NK-cell lymphomas, cutaneous NK/T-cell lymphomas, aggressive NK cell lymphomas, or chronic lymphoproliferative disorder. However, ENKTL-NT is originally a distinguishing disease, and we rarely have difficulty with a differential diagnosis.

## 10. Staging

Computed tomography (CT) and magnetic resonance imaging (MRI) scans are used for the assessment of local lesions, the involvement of the lymph nodes, and distal metastasis. In addition, a bone marrow biopsy and a gastric fiberscope are needed for the evaluation of bone marrow and gastric involvement, respectively. According to these findings, stage stratification has been performed by the Ann Arbor classification. Currently, instead of the Ann Arbor classification, the Lugano classification is recommended for the staging system [79]. Because ENKTL-NT is thought to be an FDG-avid nodal lymphoma, PET/CT imaging is also required for the Lugano classification [79].

The Ann Arbor classification subdivides patients according to the absence (A) or presence (B) of disease-related symptoms, such as prolonged fever. As mentioned above, half of the patients with ENKTL-NT had prolonged fever and were subcategorized into the presence (B) category. However, this subdividing is excluded, except for Hodgkin lymphoma, in the Lugano classification [79]. In addition, in the Lugano classification, a bone marrow biopsy is no longer indicated for the routine staging of Hodgkin lymphoma and most diffuse, large B-cell lymphomas [79]. However, in other lymphomas, a conventional biopsy is allowed because of inadequate evidence [79].

## 11. Monitoring by Using EBV Infection

Circulating EBV DNA levels measured by RT-PCR are known as a sensitive tumor marker of EBV-associated malignancies, such as nasopharyngeal carcinoma [80]. Therefore, we measured the serum EBV DNA levels of BamHI W fragments and LMP1 in 20 patients with ENKTL-NT by quantitative real-time PCR [19]. Serum EBV DNA levels were detected at high levels in all patients, but the levels were under the limit of detection in all healthy controls. The levels decreased according to the treatment and increased at relapse. The Kaplan–Meier method and univariate analyses revealed that high DNA levels of BamHI W and LMP1 at pre-treatment and high BamHI W DNA levels at post-treatment were associated with short disease-free survival and overall survival. After our report, Suzuki, et al. [81] examined the relationship between pre-treatment plasma EBV DNA levels and several clinical factors, and showed that detectable plasma EBV DNA was associated with a higher clinical stage, the presence of B symptoms, worse performance status, and higher serum soluble IL-2 receptor levels. Moreover, they showed the clinical stage and pre-treatment plasma EBV DNA were significant prognostic factors by multivariate analysis. These data suggest that the periodic measurement of serum levels of EBV DNAs is useful for diagnosis, disease monitoring, and the prediction of prognosis.

Micro-RNAs (miR) are small noncoding RNAs that control gene expression [82], and some miRs are known to be released from the cells [83]. EBV has the ability to encode circulating viral miR. We [84] examined the availability of the serum EBV miR levels as biomarkers for ENKTL-NT. Accordingly, the serum levels of miR-BART2-5p, miR-BART7-3p, miR-BART13-3p, and miR-BART1-5p were higher in patients with ENKTL-NT and significantly decreased after treatment. Moreover, a high miR-BART2-5p level correlated with a poor prognosis. Thus, circulating EBV miRs, particularly miR-BART2-5p, may be another candidate for useful diagnostic and prognostic biomarkers in ENKTL-NT patients.

## 12. Proliferation and Invasion Factors

Since the establishment of two EBV-positive ENKTL-NT cell lines SNK-6 and SNT-8 by Nagata et al. [13], the foundational investigation of ENKTL-NT has been developing rapidly. We [85,86,87] performed comprehensive array analyses in order to examine the gene expression patterns of these ENKTL-NT cell lines. Accordingly, several interesting molecules, such as the intracellular/cell surface molecule, cytokines, chemokines, and micro RNAs, were up- or downregulated, and some molecules were directly related to the lymphoma proliferation and invasion by additional in vivo and in vitro examination. These findings are schematically presented in Figure 3.

For example, IL-9 [85], soluble intercellular adhesion molecule-1 (sICAM) -1 [88], and hepatocyte growth factor (HGF) [89] were overproduced by SNK-6 and SNT-8 cells, and act as an autocrine growth factor. On the other hand, interferon-gamma-inducible protein-10 (IP-10) served as an autocrine invasion factor [86]. Cyclin-dependent kinase (CDK)1 and survivin were highly expressed in the cells and transmitted cell proliferation signals [90]. Conversely, micro RNA (miR)-15a was less expressed in the cells, and reduced antiproliferative signals [87]. In addition, other molecules were indirectly related to lymphoma proliferation in a paracrine manner. For example, hyper-produced IL-10 [91] increased the expression of CD25 (IL-2 receptor alpha) on the cells, resulting in an increased sensibility of IL-2 provided by the surrounding cells. CD70 [92] was highly expressed in the cells and induced the proliferation signal by the binding of the soluble CD27 (CD70 ligand) from bystander cells. Over-produced endogenous CCL2 and CCL22 [93], as well as IP-10 [86], attracted monocytes, which expressed membrane-bound IL-15 and induced a proliferation signal into SNK-6 in a cell-to-cell contact manner [94].

## 13. Involvement of EBV in Proliferation and Invasion Factors

Some of these molecules were suggested to be influenced by EBV. We showed that the knockdown of LMP1 in SNK-6 and SNT-8 cells induced the downregulation of CDK1 and survivin [90]. The treatment of several CDK1 and survivin inhibitors inhibited cell proliferation of the cells in a dose-dependent manner. Moreover, the Sp1 inhibitor mithramycin, one of the CDK1 and survivin inhibitors, significantly suppressed the growth of established ENKTL-NT in a murine xenograft model. On the other hand, the knockdown induced the upregulation of miR-15a in SNK-6 and SNT-8 cells [87]. miRs are small noncoding RNAs that inhibit gene expression by ligating target mRNAs to repress translation, and they play a role in various biological processes, including development, differentiation, apoptosis, and cell proliferation [82]. MYB and cyclin D1, which are validated targets of miR-15a, were highly expressed in the cells by quantitative PCR and Western blot analysis. The forced expression of a precursor miR-15a in the cells leads to decreased expressions, resulting in the inhibition of the G1 = S transition and cell proliferation. Because immunohistochemical studies revealed that CDK, survivin, MYB, and cyclin D1 were expressed in ENKTL-NT cells in the tissue section, these machineries may take place in vivo. These findings suggest that LMP1 plays an important role in cell proliferation via CDK1, survivin, and miR-15a.

IL-9 is a multifunctional cytokine mainly produced by activated Th2 lymphocytes [95]. SNK-6 and SNT-8 produced IL-9 and expressed IL-9 receptors on the cell surfaces [85]. An anti-IL-9 neutralizing antibody inhibited the growth of the cells, whereas recombinant human IL-9 enhanced their growth, suggesting that an autocrine loop of IL-9 was involved in the cell proliferation [85]. Importantly, IL-9 mRNA was not expressed in other EBV-negative NK-cell and T-cell lymphoma/leukemia cell lines, suggesting that EBV may be related to the IL-9 expression of SNK-6 and SNT-8 [85]. In fact, EBV infection of MT-2 cell, a human T-cell line, reportedly enhanced IL-9 mRNA expression, and IL-9 promoter-luciferase reporter assay revealed that EBER was responsible for IL-9 expression [96].

ICAM-1 is known as a classic cell adhesion molecule and a natural ligand of lymphocyte function-associated antigen-1 (LFA-1) [97]. sICAM-1 is a soluble form of ICAM-1, and IFN-γ induces a release of sICAM-1 by shedding membrane-bound ICAM-1 [98]. We have already shown that serum sICAM-1 levels were higher in patients with ENKTL-NT than in patients with other lymphomas [49]. Both ICAM-1 and LFA-1 were expressed in several NK-cell lines regardless of EBV infection; however, sICAM-1 was detected in culture supernatant of only EBV-positive NK-cell lines, including SNK-6 [88]. As well as IL-9, cell proliferation assay under the treatment of sICAM-1 or anti-LFA-1 antibodies revealed that sICAM-1 increased the proliferation of SNK-6 in an autocrine manner [88]. Because LMP1 induced NF-κB-dependent IFN-γ secretion in lymphoblastoid cell lines [99], LMP1 might enhance sICAM-1 release via IFN-γ-induced proteolytic cleavage. This hypothesis is supported by the finding that SNK-6 cells produced a large amount of IFN-γ [91], and that serum sICAM-1 levels were higher in patients with LMP1-positive ENKTL-NT than in those with LMP1-negative.

IP-10 is a chemokine that attracts human monocytes, activated T-cells, and NK cells expressing CXCR3 on the cell surface [100]. SNK-6 and SNT-8 produced IP-10 and expressed CXCR3 [86]. The treatment of anti-IP-10 neutralizing antibodies and recombinant IP-10 affected the cell invasion, and this showed that an autocrine loop of IP-10 was involved in the cell invasion [86]. The treatment did not affect the cell proliferation; however, surrounding monocytes enhanced the proliferation of SNK-6 and SNT-8 cells in a cell-to-cell contact manner [94], suggesting that IP-10 also took a part in the cell proliferation, indirectly. IP-10 was not produced by EBV-negative NK-cell lines, and Vockerodt et al. [101] showed that LMP1 was sufficient for inducing IP-10 expression in an examination of LMP1-transfected Burkitt’s and Hodgkin’s lymphoma cell lines.

## 14. Environmental Factors Affecting EBV Status

LMP1 is known to be important for the EBV-mediated transformation of B-lymphocytes [102]. Moreover, LMP1 acts as an oncoprotein because it can induce the transformation of rodent fibroblast cell lines [103]. According to our previous findings, as described above [85,86,87,88,90,94], LMP1 is thought to have a pivotal role in the lymphomagenesis of ENKTL-NT as well. At the point of regulation of LMP1 expression, EBNA-2 enhanced the expression by activating the LMP1 promotor in the type III latency cells, such as EBV immortalized B-lymphocytes [104]; however, the regulation system had unclear type II latency cell levels, which did not express EBNA-2. Recently, Kis, et al. [105] showed that the external stimuli, such as CD40-ligand and IL-4, could induce LMP1 in a Hodgkin lymphoma-derived cell line infected with EBV, without the expression of EBNA-2. Therefore, we examined whether LMP1 expression in SNK-6 was affected by the external stimuli. Accordingly, IFN-γ, IL-2, IL-4, IL-10, and IL-15 increased the LMP1 expression without the induction of EBNA-2 [91]. In these cytokines, IL-10 enhanced the LMP1 expression the most strongly and quickly. IFN-γ and IL-10 were detected in the supernatant of SNK-6 culture, and the treatment of blocking antibodies against these cytokines showed the downregulation of LMP1 expression. These findings suggest that IFN-γ and IL-10 sustained the LMP1 expression of SNK-6 in an autocrine manner.

The external stimuli enhancing the LMP1 expression in ENKTL-NT cells are not only these cytokines. We co-cultured SNK-6 with granulocytes and monocytes and examined whether proliferation and the LMP1 expression of the cells changed. The proliferation and LMP1 expression of SNK-6 were enhanced by co-cultured monocytes but not by granulocytes [94]. On the other hand, these enhancements were not found when monocytes were placed in a separate chamber, suggesting that cell-to-cell contact was required for these behaviors. As a key surface molecule responsible for these behaviors, we focused on membrane-bound IL-15, which is known to express in the monocytes [106] because exogenous IL-15 enhanced the proliferation and LMP1 expression of SNK-6, as described above [91], and monocytic cells reportedly activated peripheral blood NK-cells through membrane-bound IL-15 [107]. In fact, co-cultured monocytes expressed membrane-bound IL-15 on the cell surface. Moreover, the treatment of an antibody against IL-15 inhibited the monocyte-inducible proliferation and LMP1 expression of SNK-6. Immunohistochemical analysis revealed that CD14-positive monocytes preferentially colocalized with CD56-positive lymphoma cells, and therefore, the interaction between the surrounding monocytes and the ENKTL-NT cells may take place in vivo.

## 15. Therapy for Early Stage Extranodal NK/T-Cell Lymphoma, Nasal Type

For early-stage ENKTL-NT, DeVIC chemotherapy (dexamethasone, etoposide, ifosfamide, and carboplatin), concomitant with local radiotherapy (RT-2/3DeVIC), was conducted as a phase I/II trial (JCOG0211) in Japan and showed a good clinical outcome for ENKTL-NT [20]. A reason why ifosfamide and carboplatin are included in this regimen is that they are independent drugs of multidrug resistance (MDR) genes 1, which are expressed in ENKTL-NT cells [108]. Etoposide has the effect of avoiding the development of virus-associated hemophagocytic syndrome (VAHS) [109]. The disease prognosis was improved by the intervention of RT-2/3DeVIC, and the 2-year and 5-year overall survival (OS) rates were 78% [110] and 70% [111], respectively. According to these results, RT-2/3DeVIC is recognized as a standard therapy for early-stage ENKTL-NT in Japan [112]. Moreover, RT-2/3DeVIC is recommended in the NCCN (National Comprehensive Cancer Network) guideline as the preferred regimen of concurrent chemoradiation therapy [113]. As another candidate, the guideline also mentions CCRT-VIDL (concurrent cisplatin chemoradiation followed by etoposide, ifosfamide, dexamethasone, and l-asparaginase chemotherapy) [114] as another recommended regimen of concurrent chemoradiation therapy [113]. CCRT-VIDL achieved a 60% rate of 5-years of overall survival [114].

## 16. Therapy for Advanced Stage Extranodal NK/T-Cell Lymphoma, Nasal Type

For advanced-stage ENKTL-NT, the SMILE (steroid, methotrexate, ifosfamide, L-asparaginase, and etoposide) regimen was developed in Japan and East Asia [110]. L-asparaginase was thought to be essential in the regimen for the control of aggressive disease progression in spite of high toxicities [114]. In a phase II trial of SMILE [115] for patients who had newly diagnosed stage IV disease, complete remission (CR) and the 1-year OS rate were 45% and 55%, respectively, and superior to those of existing therapies [112]. According to these results, SMILE is recognized as a standard therapy for advanced-stage ENKTL-NT in Japan [112]. The NCCN guideline also suggests the AspaMetDex (l-asparaginase, methotrexate, and dexamethasone) regimen [116] as well as SMILE as a combination chemotherapy regimen [113]. A result of the AspaMetDex phase II study [116] showed around a 40% rate of 2-year OS.

Hematopoietic stem cell transplantation (HSCT) is another approach to treat advanced-stage NNKTL-NT. However, it is unclear which type of HSCT is the most appropriate [117]. The guidelines by the American Society for Blood and Marrow Transplantation support the use of both autologous and allogeneic HSCT for relapsed localized ENKL or as a front-line consolidation therapy for disseminated ENKTL-NT [118]. The NCCN guideline shows that HSCT is mainly suitable for the patients in remission of advanced-stage NNKTL-NT after first-line therapy, but it does not indicate which HSCT is better because of poor evidence [113].

The age restrictions of each therapy for early and advanced stage ENKTL-NT are controversial. RT-2/3DeVIC and SMILE are reported to be indicated for patients under 70 years old [117]. This restriction may be referred from each clinical trial [110,119]. Hematology practical guidelines from the Society of Japan have no mention of the restriction [112]. The NCCN guideline shows that a new clinical trial or radiotherapy alone is recommended in patients unfit for chemotherapy with early-stage ENKTL-NT [113]. Although the definition of “unfit patient” is not shown [113], radiotherapy alone is thought to be realistic therapy for elderly patients with early-stage ENKTL-NT. A therapy for elderly patients with advanced-stage ENKTL-NT is less certain, and we should select better therapy that fits each patient, including the best supportive care.

## 17. Arterial Infusion Chemotherapy with Concomitant Radiotherapy

Recently, we [21] have reported a novel arterial infusion chemotherapy via a superficial temporal artery with concomitant radiotherapy for patients with early-stage ENKTL-NT. The regimen was composed of methotrexate, peplomycin, etoposide, ifosfamide, carboplatin, and prednisolone (MPVIC-P), which are independent of MDR 1, as well as a DeVIC regimen. In this report, 12 Japanese patients with stage I–II were enrolled, and all patients achieved complete remission and survived without relapse. Detailed therapeutic protocols and outcomes were described previously [21].

At present, 18 patients underwent this therapy [120]. The patients’ information is summarized in Table 2. The patients’ ages ranged from 21 to 79 years (a median of 64 years), and the number of males and females were 16 and 2, respectively. There were 6 patients who had systemic symptoms, including fevers, night sweats, and weight loss. Serum lactase dehydrogenase levels and serum soluble IL-2 receptor levels ranged from 143 to 626 IU/L, with a median level of 193 IU/L, and from 237 to 990 U/mL, with a median level of 420 U/mL, respectively. In 13 (72%) of the 18 patients, serum EBV DNA copy numbers (100–270,000 copies/mL) had been detected; the levels in these 13 patients were undetectable after the therapy. With regard to the course of therapy, one patient died with systemic relapse 30 months after the therapy, and one patient survived but suffered a relapse in the larynx 13 months after the therapy. The patient underwent intravenous MPVIC-P chemotherapy as second-line therapy after the relapse, resulting in the disappearance of relapsed disease. The remaining 16 patients survived without relapse during the observation period from 26 to 111 months after the therapy (median: 73 months). For all 18 patients, the 5-year overall and disease-free survival rates are 94% and 89%, respectively. Thus, intra-arterial infusion MPVIC-P chemoradiotherapy is an effective therapy for early-stage ENKTL-NT.

## 18. Prospective Therapies

Understanding the proliferation signals in ENKTL-NT cells may make it possible to develop new therapies. Simvastatin, an inhibitor of HMG CoA reductase, is known to block the binding of ICAM-1 to LFA-1. In fact, we confirmed that simvastatin reduced the number of viable SNK-6 cells in vivo [88]. Mithramycin, an antibiotic with anti-tumor properties, downregulates both CDK1 and survivin. We confirmed that mithramycin significantly suppressed the growth of established ENKTL-NT in a murine xenograft model [90].

Highly expressed CD70 in SNK-6 [92] can be a target of immunotherapy. We have shown that the anti-CD70 antibody mediated the effective complement-dependent killing of SNK-6 [92]. The antibody may have an additional effect by inhibiting the proliferation signal mediated by the CD27–CD70 interaction. The anti-CCR4 antibody mogamulizumab has antitumor activity against cutaneous T-cell lymphoma by antibody-dependent cell killing [121], and has been already applied in a clinical setting. We [93] have shown that CCR4 was expressed in the SNK-6 and ENKTL-NT cells in the tissues section. The anti-CCR4 antibody may be useful for therapy of ENKTL-NT as well as cutaneous T-cell lymphoma.

Programmed cell death-1 (PD-1) inhibitors elicit tumor inhibitory effects by the reduction of negative immunoregulating activity through the inhibition of the attachment of PD-1 to PD-L1. We have already used some, such as Nivolumab and Pembrolizumab, against head and neck cancer in health insurance treatment. Because we found the expression of PD-L1 on ENKTL-NT cells in the tissues section [122], the inhibitors might have the same effect on ENKTL-NT as well as head and neck cancer. In fact, Kwong et al. reported that pembrolizumab was highly effective for a patient with refractory ENKTL-NT [123].

HGF is an autocrine growth factor of SNK-6, as described above [89], and its receptor, c-Met, is known as a tumor-associated antigen (TAA) for CD8^+^ cytotoxic T-cells (CTLs) [124]. Because CD4^+^ Helper T-cells (HTLs) are important for the induction of efficacious antitumor immunity [125], we examined whether c-Met on SNK-6 acts as a TAA for HTLs as well as CTLs. C-Met contained several epitope peptides, which could induce various HLA-DR-restricted specific HTLs, and these peptide-induced HTL lines have a cytolytic ability to SNK-6 [89]. In addition, we found that c-Met inhibitor ARQ197 enhanced HTL recognition by decreasing the TGF-β production by SNK-6. These results suggest that the combination of c-Met-targeted therapy and immunotherapy is a promising therapy for ENKTL-NT.

## 19. Prospective Therapies by Targeting EBV

Because EBV-related malignancies express non-self viral antigens recognized by immune cells [126], a peptide vaccine is a prospective therapy for ENKTL-NT. Demachi-Okamura et al. [127] made LMP1-specific CTLs from a healthy donor by using 43-amino acid N-terminal deletion mutant LMP1 (DeltaLMP1)-expressing APC. The CTL clone recognized a peptide of LMP1 presented by HLA-A*0206 molecules. An EBV-infected NK cell line derived from a patient with chronic active EBV infection (CAEBV) was specifically lysed by the CTL.

We [128] previously found an epitope peptide, which could bind to promiscuous MHC Class II (HLA-DR9, HLA-DR53, or HLA-DR15), by a computer-based peptide algorithm from LMP1. This peptide was naturally processed and expressed in EBV-positive NK-cell lines including SNK-6 and could elicit peptide-specific HTL, which displayed the Th1 phenotype and cytotoxic activity against the cells. Because this LMP1 epitope peptide overlaps with an HLA-A2-restricted CTL epitope, this peptide might have the ability to simultaneously induce antitumor CTL and HTL cells against ENKTL-NT cells.

LMP2A could also be a target of immunotherapy in ENKTL-NT. Although ENKTL-NT cells do not express conventional LMP2A proteins and transcripts, Fox et al. [129] reported that novel LMP2 mRNA initiated from within the EBV terminal repeats was expressed in EBV-positive NK cell lines and that LMP2-specific CTLs recognized and killed the cells. The novel LMP2 mRNA was also detected in ENKTL-NT biopsy samples. Overall, these data suggest that immunotherapy targeting LMP against ENKTL-NT may serve as an alternative therapeutic modality.

## 20. Conclusions

We described the clinical picture, diagnosis, therapy, and future prospects taken from basic and translational research studies from the standpoint of ENKTL-NT as an EBV-related lymphoma. We are optimizing EBV for the diagnosis and monitoring of ENKTL-NT. Although the therapeutic approach for ENKTL-NT has improved, the outcome, especially in advanced stages, is still unsatisfactory. Therefore, the prospective treatments mentioned above, including EBV-targeting therapy, should be developed to the next stage for clinical use, and successive research studies are also required for the discovery of new treatment strategies. We believe that further investigation will allow for ENKTL-NT to be a curable disease.

## Figures and Tables

**Figure 1 microorganisms-09-01381-f001:**
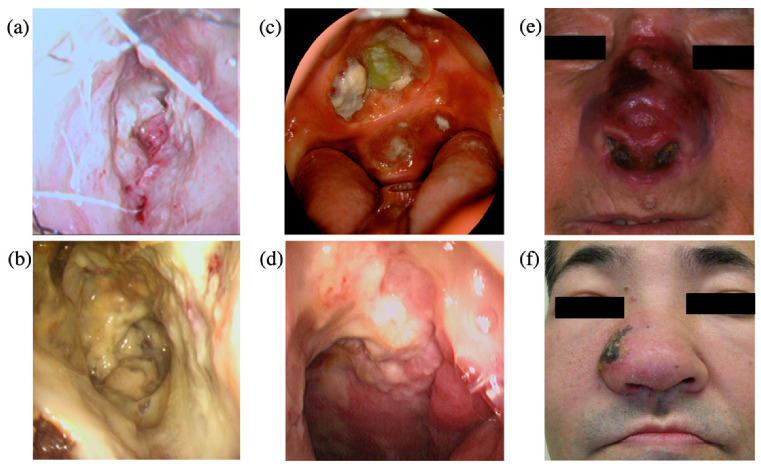
The representative local findings of extranodal NK/T-cell lymphoma, nasal type. (**a**) Granulation in the nasal cavity; (**b**) necrotic tissue in the nasal cavity; (**c**) ulceration of the hard palate; (**d**) necrotic granulation in the nasopharynx; (**e**,**f**) infiltration of the nasal skin.

**Figure 2 microorganisms-09-01381-f002:**
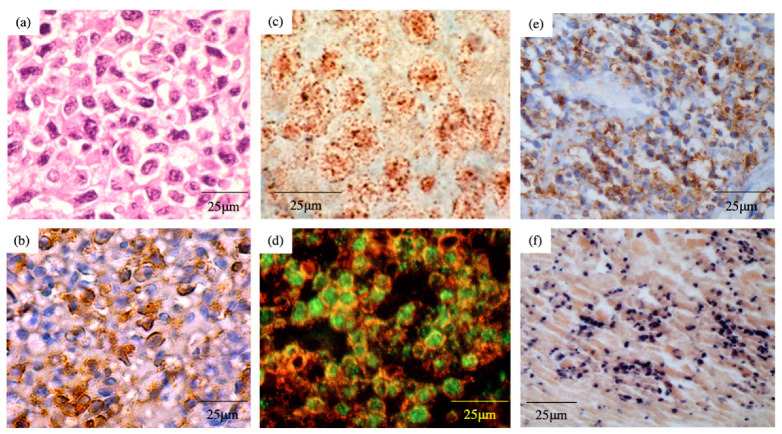
Pathological findings of extranodal NK/T-cell lymphoma, nasal type. (**a**) Hematoxylin and eosin staining; (**b**) immunohistochemical staining of CD56; (**c**) in situ hybridization of EBV DNA; (**d**) double fluorescence staining with CD2 (red) and EBV-encoded nuclear antigen (EBNA) 1 (green); (**e**) immunohistochemical staining of EBV-encoded latent membrane protein (LMP) 1; (**f**) In situ hybridization of EBV-encoded small nuclear early region (EBER) 1.

**Figure 3 microorganisms-09-01381-f003:**
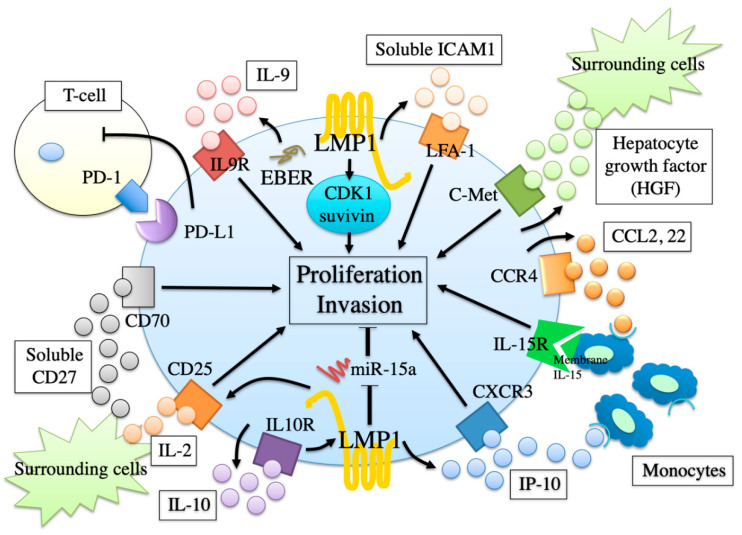
The biological characteristics of extranodal NK/T-cell lymphoma, nasal type cells. A wide variety of intracellular/cell surface molecules, cytokines, chemokines, and micro RNAs were involved in the lymphomagenesis of ENKTL-NT.

**Table 1 microorganisms-09-01381-t001:** Clinicopathological features of extranodal NK/T-Cell lymphoma, nasal type.

Country	Japan	Japan	China	Korea	Korea	Brazil
Year	2019	2010	2008	2006	2005	2011
Authors	Harabuchi et al. [18]	Suzuki et al. [44]	Wu et al. [45]	Lee et al. [3]	Kim et al. [46]	Gualco et al. [47]
Case number	62	123	115	262	114	122
Age						
Range (mean)	20–85 (53)	14–89 (52)			(47)	9–89 (45)
>60	22 (35%)		20 (18%)	55 (21%)	20 (18%)	
Sex						
Male/Female	43/19	81/42	78/29	170/92	72/42	85/37
Clinical stage						
I + II (%)	57 (92%)	84 (68%)	87 (76%)	200 (76%)	114 (100%)	25 (81%)
I/II/III/IV	44/13/1/4	55/29/8/31	61/26/8/12		83/31/0/0	23/2/2/4
Symptom						
Nasal obstruction	49 (70%)		84 (73%)			97 (80%)
Bloody rhinorrhea	29 (47%)		50 (44%)			
B symptom	32 (52%)	56 (46%)	57 (53%)	92 (35%)	35 (31%)	
Involved tissues						
Nasal cavity	60 (97%)	111 (90%)	115 (100%)		73 (64%)	97 (80%)
Hard plate	11 (18%)		8 (7%)		15 (13%)	
Facial skin	13 (21%)	19 (15%)				
Pharynx	13 (21%)	28 (23%)	27 (23%)		21 (18%)	
Lymph nodes	10 (16%)	31 (25%)	21 (18%)			
Skin	9 (15%)					
Lung/Liver	10 (16%)	10 (8%)		4 (2%)		
Digestive tracts	5 (8%)			10 (4%)		
Bone marrow	3 (5%)	9 (7%)	3 (3%)	16 (6%)		
Pathologic findings (Positive/Total cases)						
CD3	25/47 (53%)	68/86 (79%)	105/108 (97%)		104 (98%)	116/122 (95%)
CD43	31/35 (89%)	15/17 (88%)				
CD45RO	25/35 (71%)	44/49 (90%)	103/110 (94%)		61/62 (98%)	
CD20	0/59 (0%)	1/14 (7%)	0/115 (0%)		0/106 (0%)	
CD56	61/62 (98%)	115/120 (96%)	95/105 (91%)	262 (100%)	94/106 (89%)	103/122 (84%)
CD16	5/11 (45%)	9/40 (23%)				
EBER	59/62 (95%)	93/94 (99%)	106/110 (96%)	262 (100%)	46/61 (75%)	74/74 (100%)
LMP1	25/53 (47%)					10/122 (8%)
Gene rearrangement (Positive/Total cases)						
B cell receptor	0/34 (0%)					
T cell receptor	12/34 (35%)					7/74 (10%)

**Table 2 microorganisms-09-01381-t002:** Overview of the 18 patients with extranodal NK/T-cell lymphoma, nasal type treated with arterial infusion MPVIC-P chemoradiotherapy.

Case No	Age	Gender	Systemic Symtom	Performance Status	Clinical Stage	Lesion	LDH (IU/L)	sIL-2R (U/mL)	Radiation (Gy)	Response	EBV-DNA (Copy/mL)		Observation Period (Months)	Outcome
											Before Treatment	After Treatment		
1	48	Female	+	1	I	NC	205	346	56	CR	391	<100	111	Disease free
2	60	Male	-	0	I	NC	236	345	56	CR	149	<100	107	Disease free
3	64	Male	-	0	I	NC	162	290	54	CR	120	<100	107	Disease free
4	48	Male	+	1	I	NC	176	447	54	CR	1640	<100	103	Disease free
5	40	Female	+	1	I	NC	144	604	54	CR	160	<100	100	Disease free
6	70	Male	-	0	I	NC	152	528	54	CR	100	<100	89	Disease free
7	21	Male	+	0	I	NC	177	529	54	CR	100	<100	73	Disease free
8	63	Male	-	0	I	NC	151	530	54	CR	<100	<100	72	Disease free
9	58	Male	-	0	II	NC	765	2410	54	CR	270,000	<100	68	Disease free
10	47	Male	-	0	I	NC	164	298	54	CR	<100	<100	48	Disease free
11	67	Male	+	0	I	NC	626	990	54	CR	62000	<100	42	Disease free
12	21	Male	+	0	II	NC	205	406	54	CR	550	<100	39	Disease free
13	67	Male	-	0	I	NC	293	580	54	CR	790	<100	84	Disease free
14	79	Male	-	0	I	NC	193	452	54	CR	<100	<100	30	Died with disease
15	68	Male	-	0	I	NC	143	237	54	CR	150	<100	48	Disease free
16	71	Male	-	0	I	NC	144	268	54	CR	<100	<100	40	Alive with disease
17	79	Male	-	0	I	NC	168	633	54	CR	<100	<100	36	Disease free
18	66	Male	-	0	I	NC	209	420	54	CR	450	<100	26	Disease free

NC: Nasal Cavity, CR: Complete Remission.

## Data Availability

Data sharing is not applicable to this article.

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
