# Peer review of "Extranodal NK/T-Cell Lymphoma, Nasal Type: Genetic, Biologic, and Clinical Aspects with a Central Focus on Epstein–Barr Virus Relation"

_microorganisms, 2021, doi:10.3390/microorganisms9071381_

Round 1

Reviewer 1 Report

Dear Authors, thank you for having provided all the suggested  corrections to the text. I think that it is now complete. Congratulations. 

Reviewer 2 Report

Thank you for improving the paper.

This manuscript is a resubmission of an earlier submission. The following is a list of the peer review reports and author responses from that submission.

Round 1

Reviewer 1 Report

The review is exaustive regarding the pathogenetic aspects of NNKTL and the added value is due to the long experience of the research group presenting the manuscript on the disease and EBV-related lymphomatous transformation.

Major comments

- The manuscript needs an extensive english revision, with particular regard to tenses of verbes.

- Please use the term "Extranodal NK/T-Cell Lymphoma, nasal type" to define the diseae according to the WHO classification, instead of NNKTL.

- Page 2 Paragraph 3 (Epidemiology): The Authors shoud detail the incidence of NNKTL in the various geographical areas as cases per 100,000 inhabitants per year.

- Page 3 Paragraph 4 (Clinical features): The Authors should explain how the diagnosis and the staging are performed.

Are there some differential diagnosis to be excluded (This aspect could be also added to Paragraph 9)?

How is the staging performed?

Is there a role for the PET/CT scan in the staging of this disease or is the CT sufficient?

Please, refer to the Lugano classification and staging of lymphoma other than to the Ann Arbor staging system.

Please, specify that NNKTL has to be classified as an extra-nodal lymphoma in the majority of cases.

- Page 8 row 33: and 5-year OS rate is 94%. -> Please, report the 5-yrs OS rate of the whole population of 18 patients instead of that of the survived 16 patients.

- Please, specify whether the chemotherapic regimens described for the limited stage and advanced stage are age-independed or it should be considered a cut-off (Yamaguchi et al. Blood 2018 considered a cut-off of 70 years for the proposed schedules). Please, discuss the therapy for elderly patients (over 70). Another option could be to consider the fitness status of the patient to select the therapy (fit versus unfit), as indicated by the NCCN guide lines 2021.

- Please, discuss the role of hematopoietic stem cell transplant as consolidation treatment in front-line or for the recurrent disease.

- Please, compare the chemotherapic regimens with that reported in the NCCN guide lines 2021.

Minor comments

Linguistic revisions

Pag 1 row 16: were mixed with inflammatory cell -> are mixed with inflammatory cells

Pag 1 row 16: diagnosis of NNKTL required -> diagnosis of NNKTL requires

Pag 1 row 21: molecule -> molecules

Page 1 row 21: micro RNAs were -> micro RNAs are

Page 1 row 23: introduce -> introduces

Pag 1 row 31: patients with NNKTL is commonly -> patients with NNKTL are commonly

Pag 1 row 33: composes of -> is composed of

Pag 1 row 43: sensitized -> sensitive

Pag 2 row 2: NNKTL develop rapidly -> has develop rapidly

Pag 2 row 3: and micro RNAs were involved -> and micro RNAs are involved

Page 2 row 13: polymerphic reticulosis -> polymorphic reticulosis

Page 2 row 23: Chaina -> China

Page 2 row 47: have shown -> has shown

Page 3 row 2: The peak of the lymphoma was -> The peak of the lymphoma is

Page 1 row 32: not -> less frequently

Reviewer 2 Report

This is an interesting review about nasal natural killer (NK) /T-cell lymphoma. The authors analyzed genetic, biologic and clinical aspects of the disease.

The paper is well written. However, some issues remain.

If disposable, please add data about Disease Free Survival with Overall Survival. Moreover, it may be interesting to report treatments for recurrent disease or second line therapies.

Since this is a review of the literature, the authors should not report personal unpublished data (chapter 13). It is better to report data on 12 patients from reference n.21.

Chapters on therapy must be moved after those about molecular mechanisms.

Some errors and typos are present in the paper.